# Ride Your Data: Raise your Arms, Scream, and Experience your Data from a Roller Coaster Cart

Vincent Casamayou, Yvonne Jansen, Pierre Dragicevic, Arnaud Prouzeau

**Abstract**— Traditionally, data visualisation has mostly focused on finding visual encodings most suitable for two-dimensional displays and static visual examination tasks. With the emergence of Immersive Analytics, researchers explored the use of physical navigation in 3D spaces and more embodied techniques to interact with data. Yet the sense-making process to explore them has remained more or less the same to the one used on 2D screens. In this paper, we propose to take advantages of immersive technology to propose a more subjective experience of data. We present *Ride Your Data* (RYD), an immersive experience in which people ride a virtual roller coasters whose shape is dictated by data, allowing them to literally ride on their data, with climbs and drops corresponding to increases and decreases in data values. After presenting the design of RYD, we present and discuss three examples of virtual roller coasters made from real datasets. We then discuss research challenges behind the design of such experiences, and more generally, data-driven immersive experiences.

**Index Terms**—Roller Coaster, Virtual Reality, Data Visualisation, Data-Driven Experience

✦

## 1 INTRODUCTION

From time immemorial, humans have explored data to make sense of the world, using a diverse set of strategies.[1] However, research in data visualisation has mostly focused on using visual encodings optimized for two-dimensional displays and static visual examination tasks. This could change with the popularisation of virtual reality (VR) and augmented reality (AR) headsets, which spurred a new field called Immersive Analytics [28], and whose goal is to study how immersive technologies can help people make sense of data. Three-dimensional immersive visualisations open up new ways of exploring data that can exploit spatial navigation metaphors, as well as interaction techniques that are embodied [15] in the sense of using body gestures that are richer than traditional desktop interfaces and that better mimic interactions with physical objects. For example, with the Imaxes system [14], people can manipulate virtual axes as they would manipulate physical ones and position them in space to create different visualisations.

Despite the innovations brought by the area of Immersive Analytics, most immersive visualisations explored so far remain very similar to traditional visualisations in how people absorb and experience data: data is essentially transformed into an object that people are invited to visually examine and (through dynamic queries and other interactions) tune and change. Immersive visualisations do employ visual representations that are typically more complex (e.g., 3D instead of 2D) and often support more elaborate visual examination processes (e.g., involving richer body gestures or virtual locomotion), but the sense-making process remains essentially unchanged. In contrast, immersive displays have been exploited outside visualisation to elicit a range of rich, elaborate, and sometimes outright bizarre subjective experiences. For example, immersive systems exist that allow people to embody a cow and eat grass on all fours [6], travel through the human body on a small vessel and explore the entire digestive tract [5], experience what it is like to control a third arm [24], embody Kim Kardashian [29], and much more. The range of subjective human experiences that are possible with immersive displays is vast, but has barely been explored in a data visualisation context.

In this paper, we take a step in that direction and propose Ride

---

- *Vincent Casamayou, Yvonne Jansen, Pierre Dragicevic and Arnaud Prouzeau are with Université de Bordeaux, CNRS, Inria, LaBRI, France. E-mail: {firstname}.{lastname}@inria.fr*

*Manuscript received xx xxx. 201x; accepted xx xxx. 201x. Date of Publication xx xxx. 201x; date of current version xx xxx. 201x. For information on obtaining reprints of this article, please send e-mail to: reprints@ieee.org. Digital Object Identifier: xx.xxxx/TVCG.201x.xxxxxxx*

[1]For examples of early data artefacts, see `http://dataphys.org/list/`.

Your Data (RYD), a data-driven experience that takes advantage of VR's ability to elicit realistic kinetic experiences. Inspired by the NASDAQ roller coaster concept [22], we explore the design of virtual roller coasters whose shape is dictated by data, allowing people to literally ride on their data, with climbs and drops corresponding to increases and decreases in data values. After presenting the different design dimensions of RYD, we present and discuss three examples of virtual roller coasters made from real datasets. We then discuss research challenges behind the design of such experiences, and more generally, data-driven immersive experiences.

Our contributions are: (1) a design exploration of the concept of data-driven virtual roller coaster; (2) A discussion of the lessons learned by the authors after exploring and trying out these immersive experiences; (3) A discussion of future challenges and opportunities for data-driven immersive experiences.

## 2 RELATED WORK

In this section, we first discuss work in Immersive Analytics. While many papers have explored the use of immersive displays to visualise data (see the survey by Fonnet and Prié [16]), we focus on contributions that specifically take advantage of the egocentric view and 6 degrees of freedom tracking to push the limits of visualisation. We then discuss data physicalisation, and more specifically data sensification, a concept introduced by Hogan [18] to capture data artefacts that convey data by eliciting rich sensory user experiences.

### 2.1 Immersive Analytics

Kwon et al. [23] are among the first to really propose an immersive experience for visualisation by proposing a spherical graph layout centered on the user. Their user study suggests that it leads to better performance compared to a regular 2D layout, and it was preferred by participants. This type of layout has been subsequently used to visualise data (e.g. flows) on a spherical project of a map surrounding the user [35]. While users were static in the previous visualisations, they will have to move in most cases to explore the workspace. In their paper, Ulusoy et al. [33] compared a room-sized visualisation in which participants were immersed to a hand-sized and a table-sized visualisation that participants observed from the outside. Participants were slower at basic analytic tasks with the room-sized visualisation, but they felt more intense emotions.

Lee et al. [25] go a step further with *data visceralisation*, i.e., immersive data-driven experiences whose goal is to give a visceral sense of physical measurements and quantities. For instance, a data visceralisation of the size of the ten tallest buildings in the world would consist of a VR scene with 3D models of the buildings themselves, at a 1:1

scale. The authors discussed other examples conveying results from an athletic race, or the number of death by guns in the US [20]. Similar concepts were applied in sports analytics, by showing data from the point of view of the athlete and in a 1:1 scale. For instance, Ye et al. [34] visualised the trajectory of the shuttle in badminton game from the point of view of the player, while a similar approach was used for basketball data analysis and training [26, 27], with positive feedback from users who felt the experience was more real, more holistic, and provided a sense of being on the court. Similar visualisations were used for analyzing data from mixed-reality studies [19]: recorded participant trajectories were visualized in a way that provided context and a way for the experimenter to almost experience what participants experienced.

With RYD, we take inspiration from this body of work but push the explorations even further away from traditional data visualisation, by exploring data-driven experiences that do not involve encoding data into virtual objects meant to be inspected visually, but instead involve encoding data into virtual egocentric movements.

## 2.2 Data Physicalisation and Sensification

Since RYD simulates the experience of riding a physical roller coaster, it relates to work where data is conveyed through physical objects. A *data physicalisation* is a physical artefact whose geometry or material properties encodes data [21]. Contrary to on-screen and most VR visualisations, many physicalisations can be experienced with other senses than vision and afford rich sets of physical interactions: if they are small enough, people may be able to lift then, turn them around to inspect them, rearrange them, and manipulate them in a number of ways. For example, DataCoaster is a physical artefact that takes inspiration from bead maze toys where children move beads along physical curves, except the shape of the curve is data-driven [17]. Therefore, users can experience the data through hand movements. Large data physicalisations cannot be manipulated as easily, but other types of physical interactions become possible. For instance, a group of designers and architects created large-size 3D elevation maps representing the population of different cities [10]. Museum visitors could walk around them and even climb on them to experience their relief more fully.

While some data physicalisations merely replicate standard statistical charts, others (like the ones we just discussed) try to better tap into the opportunities of physicality. Others go even further and try to communicate data by eliciting rich and unconventional sensory experiences. Hogan [18] conceptualized such design practices as *data sensification*. Most of the time, data sensification designers use a specific metaphor that defines how data is mapped to user experiences. In Hogan's conceptualization, this experience is often mediated by physical motion. For example, "#Good vs. #Evil" [11] is a toy car race where the speed of each car is proportional to the amount of tweets with the hashtag Good or Evil. Similarly, in "The long run" [32], the shape of marble tracks is dictated by the cost of healthcare in the UK for different age groups. Both installations focus on entertaining and engaging audiences more than on conveying precise quantities. Moving to a larger installation and closer to the RYD concept, "My Life Don't Mean A Thing If It Ain't Got That Swing" [30] consists of a set of swings which visitors are invited to use, and whose rope length is proportional to the average reported life satisfaction of a country. A country with a high life satisfaction gives a swing with a longer rope, and thus a more enjoyable swinging experience.

The use of physical artefacts is a great way to engage audiences and have them discover data by experiencing it. However, scale and financial constraints make them hard to replicate and use outside exhibitions and public spaces. RYD circumvents this restriction by using VR to provide a data-driven experience that users can live from the comfort of their home or office.

## 3 DESIGN AND EXAMPLES

Ride Your Data (RYD) is a concept and system for turning quantitative data into virtual roller coasters that people can ride. In this section, we first discuss the rationale and the general design principles behind RYD, and then discuss specific examples and implementation details.

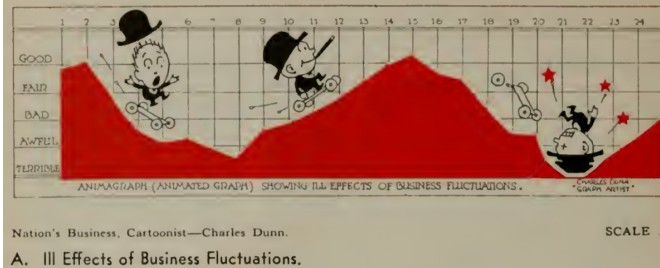

Fig. 1. Infographic of business fluctuations drawn by Charles Dunn and described in Willard Cope Brinton's book "Graphic Presentation" [8].

### 3.1 Metaphor and Sources of Inspiration

RYD taps into a metaphor of line and area charts as hills that can be climbed or descended. This metaphor is not new – for example, already in 1939, the chart specialist Willard Cope Brinton [8] commented on a business fluctuation chart showing a skate-boarder on top who uses it as a skate ramp (See Figure 1). The skate boarder is initially scared when the business value goes down as the board is hard to control; then he is happy when the value goes up, and finally crashes when the value hits the bottom of the chart. Recently, Kenny and Becker [22] employed a similar metaphor by proposing a guided tour of the Nasdaq stock market, where a 2D line chart turns into a 3D chart that the camera follows like a cart on a roller coaster. However, unlike the chart described by Brinton, there is no speed or acceleration metaphor involved, as the camera moves at constant speed. Generally, the roller coaster is a common verbal metaphor in situations where uncontrollable changes happen: for example, going through an "emotional roller coaster" is a common expression to refer to a period with a rapid succession between very different emotions. RYD takes inspiration from work by Kenny and Becker [22] and other uses of the hill metaphor, and goes further by offering people the ability to experience a *realistic* and *immersive* ride of a virtual roller coaster made from *any* dataset.

### 3.2 Motivation and Speculated Benefits

RYD is not at all meant to replace conventional data visualisations, as most analytic activities are better served by standard computer charts and visualisations, which are far easier to explore and to read. However, we believe RYD can be a useful complement in some cases. Virtual roller coasters can provide a more visceral experience of the evolution of a data value of interest and can even elicit emotional responses, especially when data values suddenly drop or climb. Although we are not aware of any study that compares the emotions elicited by virtual vs. real roller coasters, studies suggest that compared to a resting state, virtual roller coasters increase heart rate [12] and electrodermal reactions, activate brain areas involved in spatial navigation, and evoke strong spatial presence [7]. Virtual roller coasters have also been used in exposure therapy for acrophobia (i.e., fear of heights) [9]. In a data visualisation context, the emotional responses elicited by a virtual roller coaster can be useful for storytelling purposes, as they can emphasize interesting trends in the data and make the audience more engaged. For instance, if a roller coaster shows the number of COVID-19 cases over time, a COVID-19 wave would be very salient, as the sudden climb and drop would lead to a quite intense ride and thus emphasize the scale and the abruptness of the wave.

### 3.3 Design

In this section, we describe the details of the RYD design, including how roller coasters are built from data, and how the riding experience is created.

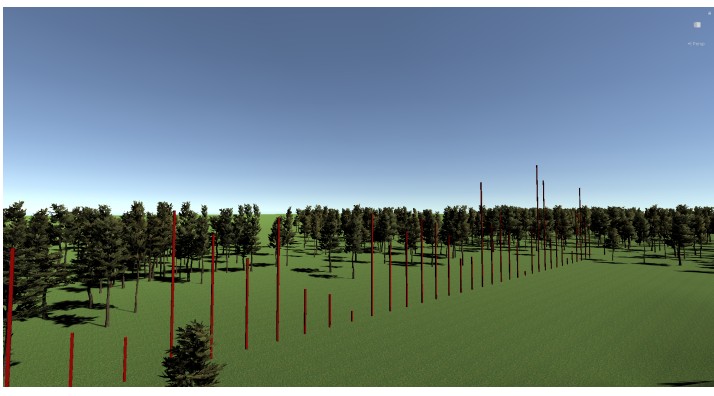
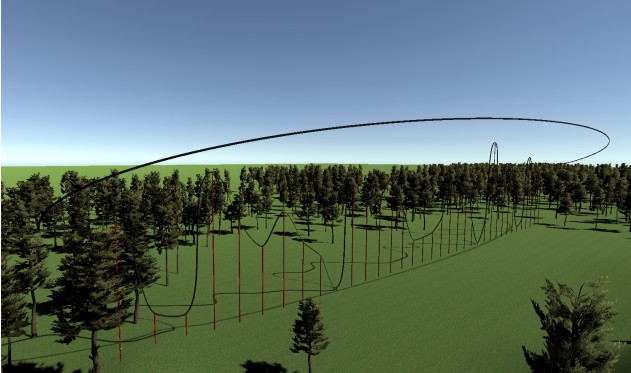

Fig. 2. *Left:* Bar chart representing the number of hospitalisations due to COVID-19 in France over time. *Right:* The resulting roller coaster.

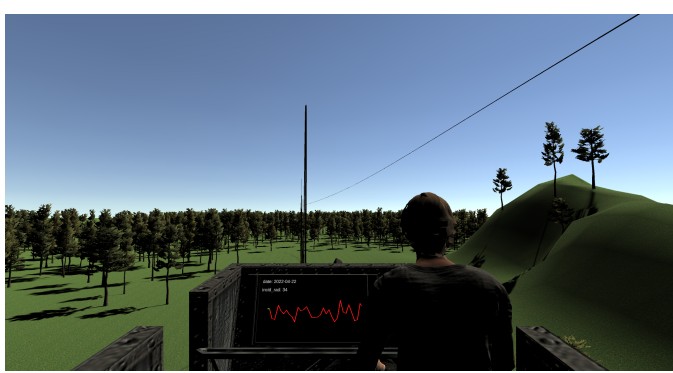

Fig. 3. RYD from the rider's perspective (COVID-19 dataset). The person sitting in front is a virtual character who simulates a riding companion.

Fig. 4. Monitor inside the cart showing detailed information during the ride.

### 3.3.1 Creating the Roller Coaster

After having identified a variable of interest in a temporal dataset[2] and selected a virtual environment to act as background, we create a bar chart that represents the evolution of the variable, and we embed it in the virtual environment (see Fig. 2-Left). We then create the track of the roller coaster by linking the top of bars using Bézier curves (see Fig. 2-Right). The use of Bézier allows the creation of a smooth track, without sharp angles. The whole roller coaster can, then, get upscaled on the X axis if the track between two points are still too sharp to be enjoyable. We keep the bars from the initial visualisation, as they stand for pillars that support the track. After the last data point in the time series, the track does a half turn and goes back to the first data point. Therefore, the whole track is a closed loop that people can ride several times. During the journey from the last data point back to the first one, riders can get an overview of the entire visualisation.

As we mentioned, we render an environment around the roller coaster, with a ground, a sky, trees, buildings, and other objects. We noticed that having clear landmarks like a ground and a sky is important to limit cybersickness and to improve realism. In future work, we consider adding more details and elements to the environment, which we expect will enhance immersion and people's aesthetic experience.

### 3.3.2 Creating the Riding Experience

In order to maximize immersion and elicit stronger emotions (see Sect. 3.2), the roller coasters in RYD are viewed using a VR headset, from the perspective of the rider who sits in a metal cart (Fig. 3). For more realism, the speed of the cart is calculated as a function of the

---

[2]In this paper we focus on one-dimensional scalar time-series, but future work should explore other types of datasets.

slope of the track, using a physics engine. This stands in contrast with the roller coaster visualisation by Kenny and Becker [22], where camera speed is constant. In RYD, the cart accelerates when the data value drops, and slows down when the data value goes up. To prevent the cart from stopping or going backwards, we set a lower bound to the speed, which simulates the traction system used in real roller coasters during steep climbs.

Because they are using a VR headset, riders can freely look around during their ride, and can physically move inside the cart. Riders can even move outside the cart if there is enough physical space around them, although admittedly, this is not something people would be naturally attempting on a real roller coaster. In addition, because of the cybersickness that can be caused by the movement and speed of the cart, we recommend riders to remain seated on a physical chair.

At the beginning of the ride, the cart is positioned on the portion of the track that does a u-turn. This allows riders to get a first overview of the visualisation before riding it, consistently with the "overview first" principle of interactive data visualisation design [31]. After all data points have been visited, the cart continues and starts a second lap. Because this time the cart does not start at zero speed, the experience riding the visualisation is a bit more intense during the second lap. This was initially not done on purpose, but we thought it was an interesting feature to have and so we kept it.

Inside the cart, a screen shows the entire visualisation as a line chart, with a dot that indicates the position of the cart (see Fig. 3 and Fig. 4). This screen acts like a minimap and provides context during the ride. The rider can get answers to potential analytical questions with more confidence, e.g., is this climb the highest one? Is the next part going to

be bumpy? This screen also displays both the *x* and *y* value of the last data point that was crossed by the cart (Fig. 4).

We added several details to enhance the experience of a ride. First, the cart rolling on the track is accompanied by a sound, which is a continuous sound whose pitch varies as a function of speed. In addition, because riding a roller coaster is not something we usually do alone, we added a virtual character sitting in front of the rider (Fig. 3), whom we named Remy. Depending on the speed and position of the cart, Remy has different reactions which he expresses with body gestures: for example, he leans forward and covers his face when the speed is really high, and applauds when a lap is over.

### 3.4 Implementation

The RYD prototype is implemented using Unity[3] and C#. The bar chart visualisations are created using IATK [13], a toolkit to create immersive visualisations on Unity. The prototype reads a CSV file with the dataset to ride and creates a bar chart. Some tweaking of the scale can be required to avoid having two local maxima too close to each other. The Bézier Path Creator plugin[4] is used to create the track made of Bézier curves. We use the physical engine provided by Unity to simulate the behavior of the cart on the track. We use the SteamVR plugin[5] to manage the integration of the VR headset in Unity. We downloaded Remy from the Mixamo[6] database along with a set of several animations. Our prototype runs smoothly on a computer with 32 Go RAM, an Nvidia Geforce GTX 1080 and an Intel Xenon W-2125. We use an HTC Vive[7] as virtual reality device.

### 3.5 Dataset Examples

To illustrate RYD, we go through the three datasets that we used to design, test, and demonstrate our system.

#### 3.5.1 COVID-19 Hospitalizations in Gironde, France

Dataset. This dataset contains multiple variables related to the evolution of COVID-19 cases in France such as the numbers of cases, hospitalized patients, patients in intensive care, and various measures of incidence rate [4]. We isolated the values between March and April 2022 (around the peak of sixth wave in France), in the Gironde department, and selected the number of new hospitalisations in a 24-hour period over time (variable = *incid_hosp*).

Roller Coaster. The roller coaster we created (shown in Fig. 2, Fig. 3 and Fig. 6a) visualises the number of new COVID-19 hospitalisations. The curve is composed of 36 points with clear inclines and drops. The data did not require any post-processing apart from the Bézier smoothing to produce a roller coaster that is enjoyable to ride.

Environment. The virtual environment surrounding the tracks is composed of trees and grass made with the Unity terrain features (see Figure 2). It is not semantically related to the data, but it helps to give a sense of scale for the tracks.

#### 3.5.2 Infant Mortality Rate in Sweden

Dataset. This dataset consists of the child mortality percentage in Sweden between 1800 and 2016 [2].

Roller Coaster. The curve produced by the dataset contains many small variations until the years 1930, where it stabilizes, making the ride really different between the first and second part. This dataset did not require post-processing either. The resulting roller coaster is visible in Fig. 5.

Environment. We chose to include a virtual environment semantically related to the data: a child bedroom (Figure 5). The roller coaster is very small in comparison to the room, making it look like a toy.

---

[3]https://wwww.unity.com
[4]https://assetstore.unity.com/packages/tools/utilities/b-zier-path-creator-136082
[5]https://store.steampowered.com/app/250820/SteamVR/
[6]https://www.mixamo.com/
[7]https://www.vive.com/

#### 3.5.3 Bitcoin Value

Dataset. This dataset consists of the evolution of the Bitcoin value between January $2^{nd}$ 2017 and November $17^{th}$ 2018, which corresponds to the period when it became popular and underwent strong variations in stock price [1].

Roller Coaster. The ride can be divided into two parts (see curve in Fig. 6b): it starts with a steep climb in price, peaks on the December $15^{th}$ 2017, then slowly declines until the final date. To produce a manageable ride, the data was smoothed using local polynomial regression fitting with a smoothing degree of 0.2, and then simplified using the Ramer-Douglas-Peucker algorithm to reduce the number of points. As a consequence, the *x* axis does not include all data points.

Environment. We chose to not include a virtual environment for this roller coaster, except a ground and a sky.

## 4 DISCUSSION

As we mentioned before, during the design process all four co-authors tried the roller coasters several times for the different datasets and discussed their impressions. From these sessions, we extracted several design considerations, research questions and potential future directions that we discuss in this section.

### 4.1 The Choice of Dataset

When designing a physical roller coaster, the engineers need to consider many parameters to identify a path that results in a thrilling ride. With RYD, the path is defined directly by the values of the dataset. Not every dataset will make for a thrilling roller coaster; a monotonic curve would be a rather boring experience more comparable to a cable car ride. With many datasets there is also a high risk that the resulting path contains drops which are too steep or has parts with many alternating climbs and drops resulting in constantly changing viewing angles which likely favor cybersickness. The Bitcoin dataset is an example demonstrating this problem. A roller coaster generated from the original, unfiltered data results in an uncomfortable experience due to many climbs and drops, as well as drops so steep that the Bezier curves invert such that the resulting path can no longer be described by a function[8]. For the Bitcoin data, we therefore chose to process the data by smoothing the curve to make the ride more enjoyable and to limit cybersickness. In the future, it would be interesting to investigate different processing methods to zero in on a balance between thrill and comfort, and see their impact on the riders' understanding of the dataset.

It might be hard for data roller coaster designers to predict how good a ride will be. Physical roller coasters are often compared by looking at properties such as the maximum g-forces they provoke, their highest drops, or the maximum speed they achieve. Games, like Roller Coaster Tycoon[9], created a rating system representing the excitement and the intensity of a roller coaster, and the potential nausea that it can provoke [3]. Exploring such type of measures would be interesting to rate the ride created with RYD to see how they are compared to real ones and see how to change them to improve the riding experience. However, while the thrill of the ride is the main criterion for a real roller coaster, a data roller coaster also needs to convey the data faithfully.

Finally, the *type* of dataset has a huge influence on the ride produced. In our three examples, we used only timeseries, which allow us to have a series of values ordered by time. These are straightforward to interpret for riders, and we are sure to have only one value for each point on the time (x) axis, which is a prerequisite to create the ride. With a more general either 2D or 3D dataset, more research is needed to see how to fit a ride to the data points that makes sense for riders. Considering additional data dimensions, would also allow the integration of loopings or inversion in the ride, which are currently not supported. What type of dataset would allow for such thrilling experience is a question worth asking.

---

[8]That is, as the curve progresses, x values decrease temporarily and the riders in the cart would find themselves head down.
[9]https://www.rollercoastertycoon.com/

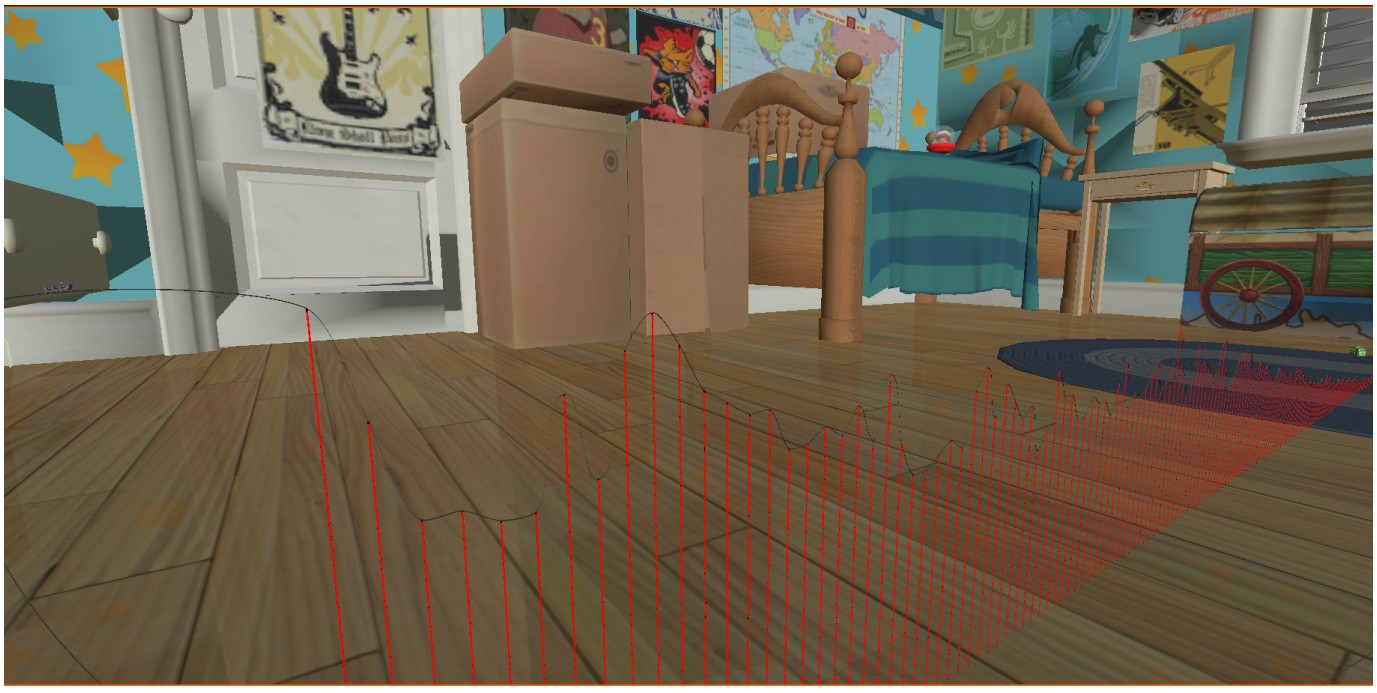

Fig. 5. Roller Coaster made with data showing the infant mortality rate in Sweden over the year in an environment representing a child bedroom.

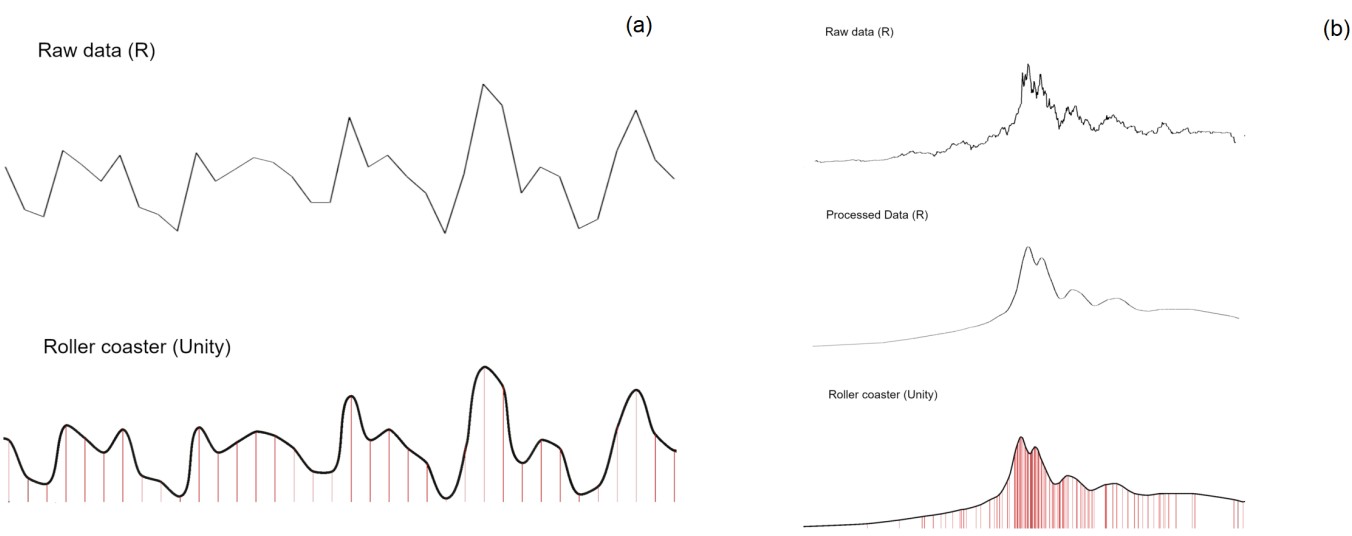

Fig. 6. **a.**Comparison between the COVID-19 cases dataset on R (upper) and the roller coaster version on Unity (lower). **b.**Comparison between the Bitcoin dataset on R without process step (upper), processed (middle) and the roller coaster on Unity (lower).

## 4.2 Integration Into a Data Analysis Process

RYD is a tool that is mainly designed as a way to tell stories with data. However, we do think that some features could be added to enable data analysis.

In a real roller coaster, there is little to no interaction with the machine and control on the ride progression. Users just sit in the cart and get carried away. In a context of data analysis, this perspective is a huge limit to the result one can benefit from. In that way, stepping back a bit from reality and allowing more freedom to the riders how they advance over the curve could enhance the experience.

In Kenny and Becker's NASDAQ coaster [22], riders are stopped at certain *events/landmarks*. A plaque is displayed telling them some relevant fact and riders have to hit a button to continue on. We could further imagine that riders gain access to a lever, that can be grabbed and manipulated with a VR controller. That lever could be used to control the speed of the cart during the ride. That way, they could fast-forward part of the dataset they are not interested in – or even go backwards – and stop on parts they want to analyse.

Another interesting context for data analysis, is the comparison and visualisation of multiple datasets at the same time. If we take as an example the child mortality in Sweden dataset that we presented earlier, riders might want to compare it with the child mortality in another country at the same period of time. RYD would then create two roller coasters – one for Sweden and one for the other country – and riders could even be allowed to jump from one to the other by interacting with a panel in the cart. This solution would probably mostly be sensible for datasets that observe major differences between them, but might be confusing for data series that are really close to each other. It also poses the question of how to perform this jump without being too disturbing for the rider.

Another approach could be to create a displayable (with a button) but not ride-able, version of a second data series, while riding the first. Like this, divergence between the two curves, even if they are close to each other would appear. The inverse problem of the previous solution could also happen here: If the two curves get really far apart, riders might lose sight of the second track and lose the ability to follow the progression. However, all these issues happen because one cannot be at two places at the same time, so why not bring some co-workers or friends with you?

In fact, collaboration is often an important part of data interpretation. With physical roller coasters, riding one is also rarely a solo experience. Usually, there is a series of two-seat carts, linked to each other, such that many people experience a ride at the same time. Yet, one rider might find a particular roller coaster extreme whereas another might find it lame. Similarly, with RYD, two riders might not share the same impression about the data, especially when talking about emotions, which are subjective experiences and not easily shareable. In general, it is easily feasible to extend RYD to let multiple people experience a data roller coaster at the same time. While a rider is even in the current version not alone – there is Remy – it is easier to exchange with real people about the ride one just had than with him. A multi-rider version of RYD could even offer new perspectives in the context of education. The aforementioned speed control lever could be controlled by a teacher who then controls the pace of the ride and can decide to stop at points of interest, and engage riders in discussions. Such an immersive data experience approach may create a more context that is more appealing to young pupils than just looking at a data curve, potentially even increasing data understanding. With the addition of a vocal chat, it could also become a remote or hybrid experience.

Coming back to our consideration about multiple datasets, shared experiences could mitigate some issues discussed earlier: Two people could ride two different datasets at the same time, exchange and then compare their experience. The result would still be rather dependent on subjective perception, but might nonetheless provide food for discussion about the ride and the data it embodies.

## 4.3 User Experience

Our work presents a first exploration of generating data-driven roller coasters from a given dataset. As mentioned in the previous section, riding roller coasters in VR likely provokes emotions and physiological responses, yet it is currently unclear how those compare to real ones and which parameters are most important to consider when designing one. Physical roller coaster have a long history and many designs have been tested (and abandoned, for example, when they proved to be too dangerous or uncomfortable for the riders). However, it would be interesting to assess formally in how far riders gain an understanding of the information contained in an dataset. For instance, in the Covid data example, how well the ride allows the rider to understand the structure of the COVID-19 wave that happened in April 2022, or in the Bitcoin example, how well it allows the rider to understand the behavior of the Bitcoin stock market price. A reflection on the methods to evaluate this is necessary.

It would also be interesting to study if the emotions provoked by the climbs and drops in the ride correlate with the emotion that one would expect from the evolution of the dataset itself. For instance, a drop can be enjoyable for the participants, and similarly, both in the Covid and child mortality data examples, drops in data values are positive events – the felt experience and the data semantics match. In the Bitcoin example, drops in the ride happen when the price of Bitcoin also drops which many probably consider as a negative event. Consequently, in this case the felt experience and data semantics do not match. However, this may also vary between riders, and some riders may find climbing up more enjoyable and do not particularly take pleasure in drops.

Finally, the surrounding of the roller coaster may have an impact on user experience. All authors agreed that the two examples having more interesting surroundings, with outside terrains or a decorated room, improved our experience and helped apprehend the structure of the ride when compared to Bitcoin example with only had a featureless gray ground and sky. However, is it necessary to have a surrounding that semantically relates to the data itself, as in the case of the childhood mortality example? Maybe in some cases, it can make the experience more memorable, but it is unclear if such an effect truly exists and if could be achieved no matter the topic of the data.

## 5 FUTURE DIRECTIONS

As we mentioned, RYD includes a virtual companion, Remy, who sits in the cart and reacts differently depending on the situation. However, his reactions are currently only a function of the cart's speed and position. It could be interesting to have Remy react to the data itself, perhaps to illustrate expected social reactions to different trends in the data, or people's average reaction to those trends. In this case, lower data values should have a positive connotation (such as lower mortality) since a falling curve is generally perceived as more pleasurable. As an example, Remy could be applauding every time COVID-19 cases go down, as opposed to only applauding at the end of a lap as is currently the case

It could be further interesting to encode data with real, physical roller coasters, which are likely to be even more engaging than virtual ones. Of course, virtual roller coasters still have many advantages over physical ones, including that they are much easier to build. In addition, one big advantage of virtual roller coasters is that users do not have to queue to ride, and that they can more easily try out many different data roller coasters in a row. However, it could definitely be interesting to envision a physical data-driven theme park (e.g., focusing on COVID-19, climate change, or human history), and study the impact of such an information medium on engagement and learning.

## 6 CONCLUSION

In this paper, we presented Ride Your Data, a VR experience in which users ride a roller coaster made with a dataset. We offered insights on how we designed and implemented a prototype and discuss various research directions based on the design and our experience riding 3 datasets. We only started exploring the design of such immersive experiences and more research is necessary to quantify our so far subjective insights regarding how we can design such rides with various datasets, its impact on rider experience, and how it can be used to do actual data analysis. We hope you enjoyed riding our paper with us and you look forward riding your data.

## CONFLICT OF INTEREST

Arnaud Prouzeau is part of the organising committee of alt.VIS 2022. He will not be involved in any stage of the reviewing process of this paper. The remaining authors have no conflicts of interest to declare.

## ACKNOWLEDGMENTS

The authors wish to thank A, B, and C. This work was supported in part by a grant from XYZ (# 12345-67890).

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
