# OpenReview forum: "Ride Your Data: Raise your Arms, Scream, and Experience your Data from a Roller Coaster Cart"
_IEEE.org/2022/Workshop/altVIS — Accept_

### Official Review · Reviewer_F3nJ · 2022-08-09

**Review:**

I like this paper. It's definitely alt. It sits exactly on the line between "completely useless" and "maybe I'd try this, maybe the emotional response you get from the roller coaster does give you some insights". Perhaps, as the authors mention in future work, it would actually be a good idea to inform and raise awareness about important topics such as climate change. I also like the idea of using the surroundings to do some more storytelling (as in the example with the child's bedroom), and it could be used to integrate panels or relevant items/characters that explain events that happened at specific times during the time series.

The main critique I have is that this would have been the perfect opportunity for a video. You can't just write a whole paper getting me all curious about riding a roller coaster of data, and then not give me at least a video. Please please please, show a demo during the presentation.

**Conflicts:**

I might have briefly met one of the authors, but I definitely don't know them personally.

**Review Inclusion:**

No

**Sufficiently Alt:**

Yes

**Superlative:**

Most nauseating. But in a good way.

---

### Official Review · Reviewer_RoYT · 2022-08-09

**Review:**

This publication walks the line between proper data physicalization and a weird idea. I would be willing to label it as sufficiently alt.

**Conflicts:**

None.

**Review Inclusion:**

No

**Sufficiently Alt:**

Yes

**Superlative:**

The most cotton candy.

---

### Official Review · Reviewer_TscZ · 2022-08-23

**Review:**

The paper discusses the design of a data-driven virtual roller coaster called Ride Your Data. The paper further discusses the challenges encountered in the design process and the future directions.

The design rationale is well explained, including the choice of background environment, the visualization, the speed of the ride, and the speed of each ride. A mini-map is also available to viewers to display the path and ups and downs of the ride.
Three different datasets were demonstrated using the proposed design. Three main takeaways were shared from experience gained throughout the process of design and implementation. I found the lesson about datasets very interesting. Without experiencing the system, it is hard to predict, but I think even a simple dataset without many ups and downs would give an interesting experience and pass the lesson of steadiness to the viewers. Although the fun part of riding will not be present!
The idea of storytelling and using the system for collaboration is appealing and rather an exciting idea to further explore if the authors decide to pursue it.

I found the last lesson about user experience difficult to measure. How can we ‘measure’ or ‘asses’ emotions to test the theories proposed? One way could be to ask users and present results qualitatively, but one needs to think deeply about the questions.
The proposed RYD design is very interesting and can be a great way to engage people in the data. I personally would be very keen to try it and experience different datasets. I am eager to see future work and studies with the proposed design. Overall, I argue for accepting this submission.


**Conflicts:**

None

**Review Inclusion:**

Yes

**Sufficiently Alt:**

Yes

**Superlative:**

Super playful

---

### Official Review · Reviewer_GkyE · 2022-08-24

**Review:**

The research is original in providing a 3-D virtual reality (VR) immersive data analysis for a variety of data as a roller coaster.
Pros:
1. the research is innovative in combinations of research questions, method, data, and application scenarios.
2. the visualization provides great user experience that focuses on egocentric view and 6 degrees of freedom tracking

Cons:
the authors can discuss less about the datasets but alternative ways to customize the process for creating the visualization that applies to other datasets.


**Conflicts:**

NA

**Review Inclusion:**

Yes

**Sufficiently Alt:**

No

**Superlative:**

Most immersive

---

### Official Review · Reviewer_hKZy · 2022-08-30

**Review:**

Meta review:

This paper was humorous and fun, but also thought-provoking and makes a contribution to the broader visualization literature on immersive data visceralization. Please refer to the reviews for suggested changes; in particular, we would all very much like to see a demo! If there is a way to facilitate participant experiences of the roller coaster during the workshop, let organizers know how best to support that effort.

Recommendation: Accept

**Conflicts:**

I am a co-organizer of alt.VIS with Arnaud Prouzeau, but we have not discussed this paper.

**Review Inclusion:**

Yes

**Sufficiently Alt:**

Yes

**Superlative:**

Most immersAHHHHHH!!!

---

### Decision · Program_Chairs · 2022-08-31

Accept